Changes in mercury content in oysters in relation to sediment and seston content in the Colombian Caribbean lagoons

Vélez-Mendoza Anubis anvelezme@unal.edu.co 1 2
Rico-Mora Jeimmy Paola 2
Campos-Campos Néstor Hernando 2
Almario-García Margui Lorena 1
Sanjuan-Muñoz Adolfo 1
1 Área de Ciencias Biológicas y Ambientales, Facultad de Ciencias Naturales e Ingeniería, Universidad de Bogotá Jorge Tadeo Lozano , Santa Marta , Colombia
2 Instituto de Estudios en Ciencias del Mar (CECIMAR), Universidad Nacional de Colombia, Sede Caribe , Santa Marta , Colombia
Zhukova Natalia
Electronic publication date: 2025 Oct 24
Publication date: 2025
Volume: 13
Electronic Location ID: e19868
Received 2025 Jan 22; Accepted 2025 Jul 17
Copyright: ©2025 Vélez-Mendoza et al.
Copyright year: 2025
Copyright holder: Vélez-Mendoza et al.
License: This is an open access article distributed under the terms of the Creative Commons Attribution License, which permits unrestricted use, distribution, reproduction and adaptation in any medium and for any purpose provided that it is properly attributed. For attribution, the original author(s), title, publication source (PeerJ) and either DOI or URL of the article must be cited.
License URL: https://creativecommons.org/licenses/by/4.0/

Keywords: Mercury, Crassostrea rhizophorae, Bioconcentration factor, Pollution index, Ages

Funding: “Redes tróficas marinas del Caribe colombiano en la era del plástico y los contaminantes tóxicos” (code MINCIENCIAS 71475)”, funded by MINCIENCIAS and Universidad Jorge Tadeo Lozano Universidad Nacional de Colombia, sede Caribe Colombia Biodiversa (I-2023) Banco de la República with the Project 5.131 from the Foundation for the Promotion of Research and Technology (Fundación para la Promoción de la Investigación y la Tecnología) The Universidad Jorge Tadeo Lozano This study was done within the research program “Redes tróficas marinas del Caribe colombiano en la era del plástico y los contaminantes tóxicos” (code MINCIENCIAS 71475)”, funded by MINCIENCIAS and Universidad Jorge Tadeo Lozano, in partnership with Universidad Nacional de Colombia, sede Caribe. The funding for Hg analysis in oysters was provided of Colombia Biodiversa (I-2023) and Banco de la República with the Project 5.131 from the Foundation for the Promotion of Research and Technology (Fundación para la Promoción de la Investigación y la Tecnología). Translation funding was provided by internal grants from the Universidad Jorge Tadeo Lozano. The funders had no role in study design, data collection and analysis, decision to publish, or preparation of the manuscript.

==============================
Total mercury (Hg) was evaluated in the mangrove oyster Crassostrea rhizophorae, in sediments and seston from the Ciénaga Grande de Santa Marta (CGSM) and Cispatá Bay (BhC) in two climatic seasons (rainy and dry). Composite samples of sediments, seston and oysters in juvenile and adult ages were collected at six stations (three in each ecosystem) and Hg was quantified by atomic absorption spectrophotometry (Environmental Protection Agency (EPA) method 7473 PLTX-017). BhC had the highest Hg concentrations in sediment, seston and oysters compared to CGSM, with values close to the tolerable threshold for the ecosystem and associated biota (TEL) of 0.13 µg/g. Although at CGSM Hg was below the TEL in sediment and was considered safe in the oyster, significant bioconcentration was evident with the metal content in the seston, indicating a potential risk to the ecosystem and humans. The variables organic matter and temperature influenced metal availability in the sediment and seston, respectively; in contrast, they had no significant relationship in the oyster. In CGSM, higher Hg concentration was recorded in adult ages, while in BhC the highest accumulation occurred in juveniles, especially during the dry season. These findings underscore the importance of continuous Hg monitoring in both ecosystems. When assessed using the adapted Nemerow Pollution Index based on the provisional tolerable weekly intake (PTWI) of one µg Hg/kg body weight, although both sites presented a very high risk in terms of consumption, they are well below the most contaminated global hotspots over the past five decades. This study also highlights the relevance of oyster age in contamination assessments, as Hg accumulation patterns vary depending on environmental and climatic conditions.

Introduction

Part of the background presented in this section was previously published as part of a preprint (Vélez-Mendoza et al., 2024; https://www.researchsquare.com/article/rs-4725392/v1). Mercury (Hg) pollution is a global environmental problem due to its ability to bioaccumulate and biomagnify in food webs, with potentially devastating effects on ecosystems (Mountouris, Voutsas & Tassios, 2002; Driscoll et al., 2013). Catastrophic events related to Hg pollution have been recorded throughout history, with the Minamata disaster in Japan being the most emblematic example. Mercury is extensively used in the gold amalgamation process, leading to its release into rivers, soils, and coastal ecosystems (Bolaños-Alvarez et al., 2024). Colombia ranks as the third country with the highest per capita mercury (Hg) emissions worldwide, with annual discharges that historically reached up to 150 tons according to Cordy et al. (2011), but have more recently been estimated between 60 and 100 tons, according to data from the University of British Columbia and the United Nations Industrial Development Organization (CNPMLTA, MADS & ONUDI, 2017), exploited by miners from the artisanal and small-scale gold mining (ASGM) (Ortega-Ramírez et al., 2023). This widespread contamination has caused serious environmental and health problems, particularly in regions like Bolivar, Antioquia, Chocó, and the Bajo Cauca. In these areas, exposure to Hg has resulted in neurological damage and renal dysfunction in local communities that consume contaminated fish (Marrugo-Negrete, Benitez & Olivero-Verbel, 2008; Alvarez et al., 2012). Additionally, contamination of major rivers such as the Atrato, Cauca and Magdalena has led to elevated Hg levels in sediments and fish, with adverse effects on biodiversity and fish populations due to reproductive and developmental toxicity (Wesche, 2021).

In the Colombian Caribbean, the presence and impact of Hg on coastal ecosystems, particularly in bivalves, has only recently gained attention. Studies conducted near Cartagena Bay and Santa Marta, in areas such as Brujas Island, Barú Island, and Taganga, have documented seasonal variations in Hg concentrations in the mangrove oyster Crassostrea rhizophorae. In Cartagena Bay, higher Hg levels were observed during the rainy season compared to the dry season. Conversely, in Santa Marta, Hg concentrations were slightly lower in the rainy season compared to the dry season (Aguirre-Rubí et al., 2017). These fluctuations are likely influenced by environmental factors such as temperature, salinity, pH, dissolved oxygen, and sediment composition, which also affect the bioconcentration process depending on the organism’s age (Cogua, Campos-Campos & Duque, 2012; Valdelamar-Villegas & Olivero-Verbel, 2018; Ucros-Rodríguez et al., 2025). Seasonal variations and environmental factors highlight the complexity of understanding Hg bioconcentration in bivalves, particularly as these factors interact with the bivalve life cycle from juvenile to adult stages (Romero-Murillo, Campos-Campos & Orrego, 2023). In addition, recent work in mangroves in Brazil and Ecuador has shown that their fine sediments favor the retention of Hg in the sediment, creating a chronic reservoir for benthic bivalves (Velásquez-López, López Sánchez & Rivera Velásquez, 2020; Rodrigues et al., 2025).

Despite the ecological and socioeconomic importance of key areas like the Ciénaga Grande de Santa Marta (CGSM) and Cispatá Bay (BhC), there is a lack of data on Hg contamination in commercially important species such as C. rhizophorae. They are also impacted by various natural and anthropogenic pressures. In CGSM, untreated wastewater and pollutants from agricultural runoff degrade water quality and increase the risk of metal contamination (Alvarez et al., 2012). Additionally, illegal mining activities upstream of the Magdalena River introduce Hg into the aquatic system (Espinosa, Parra & Villamil, 2011). In BhC, agricultural runoff, aquaculture, and urban development contribute to eutrophication, leading to the accumulation of contaminants such as Hg in sediments and biota (Marrugo-Negrete et al., 2020).

Mercury’s impact on aquatic organisms varies based on species, the form of mercury, and local environmental conditions (Richter et al., 2014). Methylmercury, due to its strong affinity in its ionic form for the thiol groups of proteins, causes negative effects even at low concentrations, disrupting the reproduction and development of aquatic species, causing detrimental effects on egg and larval formation, and leading to neurological impacts that affect behaviors such as feeding and predator avoidance (Jeong et al., 2024). Bivalves, like oysters, are especially vulnerable to Hg accumulation due to their filter-feeding behavior, which exposes them to contaminants present in the water column and sediments (Chandrasekaran et al., 2024). Which can impact their growth, reproduction, and the quality of their edible tissues for human consumption (Cardoso et al., 2009). These characteristics make bivalve’s valuable indicator species for monitoring Hg contamination in marine ecosystems (Phillips, 1977).

This study hypothesizes that Hg bioconcentration in C. rhizophorae is influenced by environmental factors, particularly the availability of Hg in seston and sediment, organism size, and seasonal climatic variations.

Materials and Methods

Study area

The Colombian Caribbean region is characterized by a bimodal climatic regime with a rainy and dry season influenced by the Intertropical Convergence Zone (ITCZ), generating periodic patterns (Restrepo & López, 2008). Trade winds predominate from December to April (dry season), changing direction to the southeast between April and November (rainy season) (Nystuen & Andrade, 1993).

The Ciénaga Grande of Santa Marta (CGSM) covers an area of 450 km2 (González-Arteaga & Ricaurte-Villota, 2023), and was declared a Ramsar Wetland and Biosphere Reserve (UNESCO, 2001), Consiting of interconnected lagoons and a sandbar to the northeast separating it from the Caribbean Sea (Restrepo Martínez, 2004; Fig. 1A). The exchange of fresh and brackish water supports the development of Rhizophora mangle (red mangrove), providing substrate for the mangrove oyster (Crassostrea rhizophorae) (Rodríguez-Rodríguez et al., 2018). CGSM is a productive tropical ecosystem, yielding significant commercial catches of fish, crustaceans, and mollusks (Sánchez-Núñez, Viloria Maestre & Rueda, 2024).

Figure 1 Study area.

(A) Location of the study sites on the Colombian Caribbean Coast, and (B) collecting oysters from mangrove roots.

Cispatá Bay (BhC), an estuarine system within the Sinú River delta, features fine to very fine sediments primarily influenced by the Sinú River. The 130 km2 estuary is predominantly covered by mangroves (Castaño, Urrego & Bernal, 2010; Fig. 1A). Rainfall averages 66 mm in the dry season and 150 mm in the rainy season, with sediment discharge increasing from 3.1 kg/day to 11.5 kg/day during the rainy period (Rangel-Ch & Arellano, 2010). BhC salinity fluctuates seasonally due to rainfall, droughts, and the mixing of fresh and brackish water.

Field phase

Oyster samples were collected from three stations in CGSM and BhC, selecting sites that represented gradients of water inflow from the sea and freshwater sources that might carry contaminants. A single sampling was conducted during each period: the rainy season (November 2021) and the dry season (March 2022). At each oyster collection site, in situ measurements of temperature, salinity, pH, and dissolved oxygen were taken at a depth of 0.5 m using WTW 3110 and YSI Pro1030 multiparameter probes. Thus, for each ecosystem (CGSM and BhC) and climatic season (dry and rainy), three data points per physicochemical variable were obtained.

277 individual oysters were collected from CGSM, and 237 from BhC under the collection permit for wild species specimens of biological diversity for non-commercial scientific research purposes, granted by the Autoridad Nacional de Licencias Ambientales (ANLA) through Resolution 1271 of October 23th 2014, modified by resolutions 1715 of December 30th 2015 and 00213 of January 28th 2021, to the University of Bogota Jorge Tadeo Lozano (UTADEO). The samples were divided into six groups per station in each climatic season, with three groups consisting of juveniles (22.0–32.0 mm) and three groups of adults (35.0–56.5 mm) (Pacheco Urpí, Cabrera Peña & Zamora Madriz, 1983; Madrigal Castro et al., 1985). The result was a total of 71 composite samples across both ecosystems and climatic periods (Table S4).

At each station, specimens were primarily collected from the roots of mangrove trees. oysters were placed in plastic containers, were cleaned to remove any particles adhering to their shells (Fig. 1B), stored in pre-labeled, airtight polyethylene bag, and preserved in coolers with gel ice packs (∼4 °C).

To determine mercury content in seston, and to serve as food for filter-feeding organisms like bivalves, three water samples were collected at each station in 2.8 L amber flasks and kept cold (∼4 °C). After homogenization, samples were filtered through two Whatman GF/C glass fiber filters (47 mm diameter) per sample using a manual vacuum pump. Filters were then stored in hermetically sealed, pre-labeled polyethylene bags, dried in an oven at 45 °C for 24 h, and weighed on an analytical balance (Cogua, Campos-Campos & Duque, 2012).

In each station, three sediment samples were collected using a van Veen dredge. From each composite sample per station, 600 g of sediment was separated for mercury analysis, 75 g for organic matter determination, 75 g for redox potential measurement, and 300 g for granulometric analysis focused on the content of silt. Samples were stored in airtight polyethylene bags using a silicone scoop to avoid contamination, ensuring no contact with the dredge edges. Samples were kept chilled (∼4 °C) (Cogua, Campos-Campos & Duque, 2012).

Laboratory phase

To determine organic matter content, 5 g of dry sediment were placed in pre-weighed porcelain crucibles, subjected to calcination in a muffle furnace at 550 °C for 5 h, and then left in a desiccator for 2 h. Organic matter content was calculated based on the difference between the dry weight and the weight after calcination (Kenny & Sotheran, 2013).

For redox potential quantification, sediment samples were dried at 40 °C for 24 h. A portion of 25 g of sediment was then homogenized in 50 mL of deionized water using a VELP Scientifica magnetic stirrer for 30 min. Redox potential was measured with a YSI Pro1030 multiparameter probe equipped with an oxidation–reduction potential electrode, at a standard temperature of 25 °C (Aldridge & Ganf, 2003).

The 300 g of collected sediments were dried in an oven at 90 ± 5 °C. From the dried material, 100 g were weighed and it’s dispersed in one L of deionized water with 40 mL of 1% sodium hexametaphosphate ((NaPO3)6, 6.2 g/L) for 24 h to facilitate particle separation, following ISO 11277:2020 (ISO, 2020). Once dispersion was achieved, the samples were sieved covering the range from 1 mm to 63 µm. Although each sediment fraction was classified according to size (very coarse sand: 1 mm; coarse sand: 500 µm; medium sand: 250 µm; fine sand: 180–125 µm; very fine sand: 90–63 µm; silt: <63 µm), only the final weight of the <63 µm fraction (silt content) was recorded and used in this study for a descriptive analysis.

For the chemical analyses, all materials were pre-treated by purging with 5% nitric acid (HNO3) and deionized water for 24 h to prevent contamination. Samples were handled with gloves, glass, or plastic materials to avoid contamination. Samples were then transferred to the Toxicology and Environmental Management Laboratory at the University of Córdoba, preserved at ∼4 °C, for mercury (Hg) quantification.

The anteroposterior length (APL) of the oyster was measured on the inner side of the ventral valve using a Vernier caliper (precision of 0.05 mm). soft tissues were removed and weighed using an analytical balance  ± 0.1 mg). For each sample, organisms of similar size were pooled, and soft tissues were placed in pre-weighed and labeled 30 mL glass vials. The vials were then lyophilized, and the final dry weight was recorded.

For mercury analysis in samples of seston, sediment, and oyster tissue underwent a pre-treatment and analysis at the Toxicology Laboratory, following a validated method. The pre-treatment included homogenizing the lyophilized samples (oyster tissue), weighing approximately 20–40 mg of each sample into nickel cells, and introducing them directly into a Direct Mercury Analyzer (DMA-80). This instrument operates based on atomic absorption spectrometry with thermal decomposition and gold amalgamation, in accordance with US Environmental Protection Agency (EPA) Method 7473 (USEPA, 2007).

During the combustion or pyrolysis step, all mercury compounds in the sample including volatile organic mercury species are thermally decomposed and converted to elemental mercury vapor (Hg°). Combustion by products, such as halogenated compounds and nitrogen and sulfur oxides, are removed by passing the carrier gas (oxygen) through a hot catalyst. The elemental mercury vapor, free from interferences, is transported by the carrier gas to a gold trap, where it is selectively adsorbed to form a gold–mercury amalgam (Milestone, 2020).

Once a sufficient amount of mercury has accumulated in the gold trap, it is rapidly heated to release the concentrated elemental mercury vapor. This vapor passes through an absorption cell in the path of an ultraviolet (UV) light beam at the mercury-specific wavelength (253.7 nm), allowing precise quantification.

Importantly, this closed system prevents any loss of mercury species including volatile organic forms ensuring accurate measurements. Therefore, there is no risk of mercury loss due to volatilization during the pre-treatment and analysis steps.

Sediment and seston samples were subjected to standard procedures for trace metal analysis in environmental matrices. The fraction smaller than 63 µm was selected, as this size fraction is known to better retain the largest amount metals. Both sediment and seston samples were digested using 5% nitric acid (HNO3) to prepare them for total Hg determination. For seston, the laboratory adapted the digestion protocol originally developed for the sediment samples, ensuring methodological consistency between matrices. Total Hg concentration was subsequently quantified following EPA Method 7473 PLTX-017, which involves thermal decomposition, amalgamation, and atomic absorption spectrometry (EPA, 1998).

For analytical control, triplicate analyses of Hg solutions at different concentrations were performed. Calibration curves were established for three concentration ranges in all matrices: 0.005–0.02 µg Hg, 0.02–0.05 µg Hg, and 0.05–0.5 µg Hg. The coefficients of determination (R2) were 0.9993, 0.9986, and 0.999 for sediment; 0.9973, 0.9962, and 0.9979 for seston; and 0.9996, 0.9996, and 0.9989 for oyster tissue. Error percentages were kept below 15% for all three matrices. TORT-1 (lobster hepatopancreas) used as a reference material from the National Research Council of Canada (NRCC) for samples of oyster tissue, and GSD-10 for sediment and seston samples (Leng et al., 2013; Romero-Murillo, Campos-Campos & Orrego, 2023). Recovery percentages were 100 ± 1.4% for sediments (limit of detection (LOD) = 0.00073 µg/g Hg), 100 ± 5.4% for seston (LOD = 0.000015 µg/g Hg), and 100 ± 1.4% for oysters (LOD = 0.00073 µg/g Hg).

Data analysis

The bioconcentration factor (BCF) was calculated as the ratio of Hg concentration in oyster tissue to its presence in sediment (sd) and seston (st), expressed in parts per million (ppm, µg/g) in dry weight (d.w.). BCF was used to evaluate the efficiency of Hg accumulation in oyster soft tissue. Mountouris, Voutsas & Tassios (2002), BCF <1 suggests no metal accumulation, BCF ≥ 1 and <10 indicates accumulation and BCF ≥ 10 indicates hyperaccumulation of metal. The calculation was based on Mountouris, Voutsas & Tassios (2002) and Romero-Murillo, Campos-Campos & Orrego (2023). (1) BCFsd=MetalorganismMetalsediment,BCFst=MetalorganismMetalseston.

Permutation analysis of variance (PERMANOVA) was applied to compare Hg concentration in oyster tissue and its BCF between the two ecosystems (k = 2), the two climatic seasons (k = 2), the six stations (k = 6) and the two categorized size classes (k = 2). 9,999 permutations were performed using Euclidean distance and type III sum of squares. p-values were computed using Monte Carlo (MC) permutation testing only when unique permutations were less than 100 (Anderson, Gorley & Clarke, 2008).

Relationships between mercury (Hg) concentrations in sediment and seston were examined using Pearson’s and Spearman’s correlation analyses based on data distribution. Normality tests (Shapiro–Wilk) were performed on each dataset prior to analysis (Zar, 2010). Influences of environmental predictor variables on the oyster Hg concentration and Hg BCF in relation to seston were evaluated using a distance-based linear model (DistLM) with adjusted R2 criterion and 9,999 permutations (Anderson, Gorley & Clarke, 2008).

To evaluate mercury contamination in bivalves, the Nemerow integral contamination index −Pc- was used (Ding et al., 2022). Calculations of Pc are based on the average value of the individual pollution index (Pavg), the maximum value (Pmax) and the minimum value (Pmin) of the collected data standardized to µg/kg wet weight (w.w.) of the bivalves. Cavg average concentration value, recorded in the data set evaluated, Cmax and Cmin are the maximum and minimum concentration values from the same data set, and S is the provisional tolerable weekly intake (PTWI) with Hg in adults (one µg/kg body weight per week) (FAO/WHO, 2010). The individual index values were calculated using the following formula: (2) Pavg=CavgS,Pmax=CmaxS,Pmin=CminS.

Once the historical values of Pavg, Pmax and Pmin were obtained for each location by year, the calculation of Pcwas performed establishing (i) Pc ≤ 0.7 considered no risk, (ii) 0.7 <Pc ≤ 1 low risk, (iii) 1 <Pc ≤ 2 medium risk, (iv) 2 <Pc ≤ 3 high risk and (v) Pc >3 very high risk of contamination (Ding et al., 2022). ΣPc is the sum of all Pc values divided by “n”, the total number of years evaluated per location in its historical record, ensuring that the values of Pmax and Pmin do not overestimate or underestimate the calculation of the contamination index for each metal evaluated, respectively. It was calculated using the following modification of the equation: (3) ∑Pcn=Pavg2+Pmax2+Pmin23.

A hierarchical clustering analysis was carried out using the squared Euclidean distance with Ward’s linkage, minimizing variability and producing uniformly sized clusters (Tudor et al., 2002; Tudor & Williams, 2004).

Results

Environmental conditions in both coastal lagoons

In CGSM, the temperature during the rainy season was 31.17 ± 0.48 °C (n = 3), while in dry season it was 30.53 ± 0.84 °C (n = 3). In BhC the temperature during the rainy season was 28.97 ± 0.43 °C (n = 3), and in dry season it was 29.75 ± 0.03 °C (n = 3) (Fig. 2A, Table S1).

Figure 2 Total Hg concentration in sediments, seston and oysters.

(A) Physicochemical variables measured in situ, and (B) Total Hg concentration in mg/g dry weight (d.w.) in sediments and seston and mean concentration (±standard error) in tissue of the oyster Crassostrea rhizophorae at each station in Ciénaga Grande de Santa Marta and Cispatá Bay during the rainy season of 2021 and dry season of 2022.

During the rainy season, CGSM had a higher pH value (8.77 ± 0.12, n = 3) compared to BhC (7.84 ± 0.09, n = 3), while during the dry season, were a decrease in pH in CGSM (8.44 ± 0.16, n = 3) and an increase in BhC (8.19 ± 0.003, n = 3) (Fig. 2A, Table S1).

The average dissolved oxygen content was higher in CGSM during both climatic seasons, with values of 7.83 ± 0.66 mg/L (n = 3) in rainy season and 7.39 ± 2.4 mg/L (n = 3) during dry season. In BhC, the contents were 4.32 ± 0.5 mg/L (n = 3) in rainy season and 5.52 ± 1.28 mg/L (n = 3) along the dry season (Fig. 2A, Table S1).

In CGSM, salinity values varied significantly between seasons, ranging from 2.47 ± 1.01 (n = 3) during the rainy season to 18.53 ± 6.33 (n = 3) in the dry season. In contrast, BhC showed less seasonal fluctuation, with average salinity levels from 24.9 ± 1.01 (n = 3) to 30.83 ± 0.56 (n = 3) across the two seasons (Fig. 2A, Table S1).

Organic matter in CGSM was also seasonally variable, with contents during the rainy season at 11.67 ± 3.27% (n = 3), approximately double the levels observed in the dry season (5.97 ± 2.4%, n = 3). BhC, on the other hand, exhibited minor seasonal differences in organic matter, ranging from 5.6 ± 0.5% (n = 3) to 6.06 ± 1.28% (n = 3) (Fig. 2A, Table S1).

In sediments the redox potential, both in CGSM and BhC, reducing conditions were recorded with a range of values from 28 to 77 mV between both sectors. Increases in redox potential were observed in BhC (52 ± 3, n = 3 to 65 ± 4, n = 3), and decreases in CGSM (50 ± 11, n = 3 to 35 ± 4, n = 3) from rainy season to dry season (Fig. 2A, Table S1).

The silt content showed marked differences between ecosystems and across climatic seasons. In CGSM, average silt content increased notably from 19.95 ± 8.68% (n = 3) during the rainy season to 44.33 ± 11.94% (n = 3) in the dry season, indicating greater fine particle accumulation in the latter period. In contrast, Cispatá Bay exhibited consistently higher silt proportions, with averages of 66.40 ± 5.76% (n = 3) in the rainy season and 66.62 ± 12.14% (n = 3) in the dry season, reflecting stable conditions favorable for fine sediment retention (Table S1).

Concentration of Hg in sediments, seston and C. rhizophorae

Hg concentration in sediments and seston varied markedly between the two ecosystems. In BhC, in both sediments and seston, Hg concentrations are consistently higher than in CGSM in both climatic seasons (Fig. 2B, Table S2).

In the rainy season, the highest concentration of Hg in sediments was found in CIS-1 (BhC) with 0.128 µg/g Hg dry weight (d.w.) which is double the highest content detected in CGSM (0.059 µg/g Hgd.w. in CGS-1). Hg content in sediments at BhC was slightly lower in the dry season but remained above 0.08 µg/g Hgd.w. indicating a possible constant source of Hg contamination. In CGSM, during the dry season, lower Hg was observed in CGS-1 and higher in CGS-2 (Fig. 2B, Table S2).

The Hg available in the seston presented similar values in the stations of each ecosystem and in the two climatic seasons. However, as in sediments, a lower concentration was detected in the dry season. In BhC, the concentration went from 0.032 ± 0.005 µg/g Hgd.w. in the rainy season to 0.022 ± 0.001 µg/g Hgd.w. in the dry season. Lower concentrations were found in CGSM, with values ranging from 0.01 ± 0.001 µg/g Hgd.w. in the rainy season to 0.004 ± 0.001 µg/g Hgd.w. in the dry season (Fig. 2B, Table S2).

In BhC the highest Hg content in seston was positively and significantly related to temperature (Pearson, r = 0.933, df = 11, p-value = 0.006) and in sediments Hg was significantly related to organic matter (Pearson, r = 0.845, df = 11, p-value = 0.039) (Fig. 2, Table S3).

Hg concentrations in oyster tissue show distinct accumulation patterns in CGSM and BhC, varying as a function of climatic seasons. However, a pattern similar to that of sediment and seston was maintained, with a higher Hg content in oyster soft tissue in BhC (Fig. 2B). In the rainy season, Hg in the oyster was 0.083 ± 0.007 µg/gd.w. (n = 17) in CGSM and of 0.135 ±0.015 µg/gd.w. (n = 18) in BhC showing significant differences between the two ecosystems (Permanova, p-value < 0.01). In dry season, these differences were maintained, given the decrease in oyster Hg content in CGSM (0.066 ± 0.007 µg/g Hgd.w., n = 18) and increased in BhC (0.154 ± 0.019 µg/g Hgd.w., n = 18) (Tables 1 and 2, Fig. 3A, Table S4).

Table 1 Mercury content and bioconcentration factors in Crassostrea rhizophorae.

Mercury concentration in dry weight (d.w.) and bioconcentration factors in tissues of juvenile and adult sizes of the oyster Crassostrea rhizophorae during the rainy and dry seasons at CGSM stations (CGS-1, CGS-2, and CGS-3) and Cispatá Bay (CIS-1, CIS-2, and CIS-3).

Age	Season	Station	APL (mm)*	Hg (µg/g d.w.)*	BCF-Hg (sediments)*	FBC-Hg (seston)*	
Juvenile	Rainy	CGS-1	29.96 ± 0.24
29.19–30.44 (20)	0.11 ± 0.01
0.09–0.12 (3)	1.84 ± 0.16
1.55–2.08 (3)	9.15 ± 0.78
7.68–10.34 (3)	
CGS-2	28.8 ± 0.34
26.22–30.38 (21)	0.06 ± 0.004**
0.05–0.06 (3)	1.38 ± 0.09
1.25–1.57 (3)	5.48 ± 0.37
4.95–6.19 (3)	
CGS-3	25.56 ± 0.35
23.88–27.81 (20)	0.08 ± 0.03
0.05–0.14 (3)	2.28 ± 0.72
1.48–3.72 (3)	9.49 ± 2.98
6.15–15.43 (3)	
CIS-1	28.33 ± 0.23
26.75–29.5 (20)	0.13 ± 0.04
0.09–0.21 (3)	1.04 ± 0.29
0.74–1.62 (3)	5.6 ± 1.55
3.97–8.7 (3)	
CIS-2	29.29 ± 0.3
28.88–30 (20)	0.12 ± 0.02
0.09–0.16 (3)	1.21 ± 0.25
0.95–1.7 (3)	3.65 ± 0.74**
2.86–5.14 (3)	
CIS-3	27.79 ± 0.3
25.88–28.88 (20)	0.23 ± 0.05***
0.12–0.28 (3)	2.79 ± 0.64
1.52–3.53 (3)	5.62 ± 1.28
3.06–7.1 (3)	
Dry	CGS-1	28.43 ± 0.22
25.69–30.46 (37)	0.06 ± 0.02
0.04–0.09 (3)	15.17 ± 3.52***
9.91–21.84 (3)	20.93 ± 4.85***
13.67–30.14 (3)	
CGS-2	27.93 ± 0.24
24–31.2 (41)	0.06 ± 0.01
0.05–0.08 (3)	0.88 ± 0.12**
0.72–1.17 (3)	15.33 ± 2.08
12.58–19.4 (3)	
CGS-3	28.21 ± 0.14
24.74–31.28 (52)	0.04 ± 0.01**
0.03–0.06 (3)	1.07 ± 0.25
0.8–1.57 (3)	7.87 ± 1.82
5.84–11.5 (3)	
CIS-1	28.2 ± 0.16
25.2–31.3 (35)	0.09 ± 0.01
0.08–0.1 (3)	0.74 ± 0.06**
0.63–0.82 (3)	3.91 ± 0.3
3.33–4.32 (3)	
CIS-2	27.7 ± 0.13
24.47–31.22 (36)	0.27 ± 0.02***
0.24–0.31 (3)	3.38 ± 0.23
3.01–3.79 (3)	12.4 ± 0.84
11.04–13.92 (3)	
CIS-3	27.51 ± 0.12
24.3–30.71 (27)	0.2 ± 0.07
0.1–0.33 (3)	2.68 ± 0.92
1.39–4.46 (3)	9.86 ± 3.39
5.14–16.43 (3)	
Adult	Rainy	CGS-1	50.75 ± 0.47
45.75–55 (10)	0.1 ± 0.01
0.08–0.11 (3)	1.68 ± 0.14
1.41–1.85 (3)	8.31 ± 0.68
6.98–9.2 (3)	
CGS-2	49.17 ± 0.56
44.75–56.75 (10)	0.09 ± 0.02
0.06–0.13 (3)	2.32 ± 0.53
1.49–3.31 (3)	9.19 ± 2.1
5.88–13.07 (3)	
CGS-3	38.81 ± 0.75
36.25–41.38 (8)	0.05 ± 0.01**
0.04–0.06 (2)	1.38 ± 0.19
1.19–1.57 (2)	5.74 ± 0.79
4.943–6.531 (2)	
CIS-1	37.96 ± 0.25
36.13–39.5 (10)	0.12 ± 0.03
0.08–0.18 (3)	0.95 ± 0.25
0.63–1.44 (3)	5.14 ± 1.34
3.39–7.77 (3)	
CIS-2	42.15 ± 0.71
40.2–45 (11)	0.09 ± 0.01
0.07–0.11 (3)	0.9 ± 0.11
0.78–1.13 (3)	2.73 ± 0.35**
2.35.42 (3)	
CIS-3	39.23 ± 0.33
37–42.2 (11)	0.13 ± 0.01
0.12–0.14 (3)	1.60 ± 0.08
1.44–1.71 (3)	3.22 ± 0.17**
2.89–3.44 (3)	
Dry	CGS-1	47.21 ± 0.36
41.38–53.5 (14)	0.05 ± 0.004**
0.05–0.06 (3)	12.16 ± 0.93***
10.88–13.97 (3)	16.78 ± 1.29
15.02–19.28 (3)	
CGS-2	44.13 ± 0.353
39.5–49 (24)	0.06 ± 0.003**
0.06–0.06 (3)	0.85 ± 0.04**
0.81–0.92 (3)	14.79 ± 0.63
14.01–16.03 (3)	
CGS-3	41.95 ± 0.32
38.56–46 (20)	0.12 ± 0.02
0.09–0.15 (3)	3.29 ± 0.42
2.56–4.03 (3)	24.18 ± 3.11***
18.81–29.57 (3)	
CIS-1	48.83 ± 0.46
44–53.75 (13)	0.10 ± 0.01
0.09–0.12 (3)	0.86 ± 0.08**
0.73–0.99 (3)	4.51 ± 0.4
3.85–5.22 (3)	
CIS-2	45.1 ± 0.34
39.8–50 (14)	0.11 ± 0.02
0.09-0.15 (3)	1.36 ± 0.23
1.05–1.82 (3)	5.01 ± 0.85
3.86–6.67 (3)	
CIS-3	38.5 ± 0.12
35.5–41.5 (20)	0.15 ± 0.04
0.09–0.23 (3)	2.02 ± 0.54
1.28–3.07 (3)	7.44 ± 1.99
4.73–11.31 (3)	
Notes.

* Here, the value of ‘n’ in parentheses refers to the total number of individuals included in each size category.

APL anteroposterior length

Maximum (***), and minimum (**) values are shown.

Table 2 PERMANOVA analysis on Hg concentration and bioconcentration factor (BCF-Hg) vs. sizes (juveniles and adults), stations, ecosystem, and climatic season in the oyster Crassostrea rhizophorae.

Factor	df	SS	MS	Pseudo-F	Únique	p -value	
Hg (µg/g d.w.)	
EC	1	8.8  × 10−2	8.8  × 10−2	47.31	9 836	1 × 10−4	
CS	1	4.6  × 10−5	4.6  × 10−5	0.03	9 837	0.876	
ST (EC)	4	2.5  × 10−2	6.2  × 10−3	3.31	9 948	1.6 ×10−2	
EC   × CS	1	5.3  × 10−3	5.3  × 10−3	2.85	9 843	0.098	
AG (ST (EC))	6	4.6  × 10−2	7.7  × 10−3	4.11	9 954	1.8 ×10−3	
CS   × ST (EC)	4	2.9  × 10−2	7.4  × 10−3	3.94	9 949	7 ×10−3	
CS   × AG (ST (EC))	6	2.5  × 10−2	4.2  × 10−3	2.27	9 939	0.049	
Residual	47	8.8  × 10−2	1.87  × 10−3				
Total	70	3.07  × 10−1					
BCF-Hg	
EC	1	747.7	747.7	72.14	9 841	1 ×10−4	
CS	1	594.8	594.8	57.39	9 864	1 ×10−4	
ST (EC)	4	62.78	15.7	1.514	9 954	0.219	
EC   × CS	1	152.9	152.9	14.76	9 837	4 ×10−4	
AG (ST (EC))	6	200.8	33.5	3.23	9 949	0.011	
CS   × ST (EC)	4	84.1	21.03	2.03	9 951	0.107	
CS   × AG (ST (EC))	6	322.4	53.7	5.18	9 931	4 ×10−4	
Res	47	487.1	10.3				
Total	70	2 720					
Notes.

p-value < 0.05 indicates significant differences among the analyzed factors. In bold the significant ones.

Figure 3 Comparative analysis of concentrations and bioconcentration factor of Hg in Crassostrea rhizophorae.

Comparative analysis of concentrations (A) and bioconcentration factor (B) of Hg in the oyster Crassostrea rhizophorae between Ciénaga Grande de Santa Marta (CGSM) and Cispatá Bay (BhC): effects of organism size and climatic season. Above the green line, Hg accumulation condition in the oyster is considered (BCF≥ 1), and above the red line, hyperaccumulation of the metal is considered (BCF≥ 10).

With respect to Hg BCF, both in sediments and seston, both ecosystems presented accumulation to hyperaccumulation of Hg in the oyster tissue, with the highest values in the dry season. In this same climatic season, at CGSM, the oyster presented an accumulation of Hg with the sediment (BCF≥1) and a hyperaccumulation of the metal with the seston (BCF≥10), as opposed to the accumulation condition in both matrices during the rainy season. In BhC, the oyster maintained the accumulation condition in both matrices (BCF≥1) as in rainy season (Tables 1 and 2). These results notably emphasize the capacity of the oyster to accumulate Hg in its tissues, especially in CGSM through the seston in the dry season. Significant differences in BCF were determined between CGSM and BhC ecosystems, with higher concentrations in BhC (Permanova, p-value < 0.05). However, no significant differences in concentrations were observed between climatic seasons, since the values measured at two of three stations in both CGSM (CGS-2 and CGS-3) and BhC (CIS-1 and CIS-3) were similar in both climatic seasons (Tables 1 and 2, Fig. 3B).

In the two ecosystems evaluated, there were significant differences in the Hg contents between stations (Permanova, p-value < 0.05; Table 2). In the rainy season at CGSM the differences occurred between stations CGS-1 and CGS-2 and at BhC between CIS-2 and CIS-3. In the dry season, significant differences were found between CIS-1 and CIS-2 in BhC, with the lowest concentrations in CIS-1 (Table 1, Fig. 2B, Tables S4 and S5).

There were significant differences in the length of juveniles and adults (Permanova, p-value < 0.05) between stations (Table 2). In CGSM in the dry season, concentrations were higher in adult ages at station CGS-3 (0.121 ± 0.016 µg/g Hgd.w.) with respect to juveniles (0.039 ± 0.009 µg/g Hgd.w.). In BhC, in both rainy and dry seasons, the highest concentration of juvenile lengths was at station CIS-2 (Table 1, Fig. 3A, Tables S4 and S5).

Importance of seston in the bioconcentration of Hg

High Hg bioconcentration factor values reflected a significant correlation with seston content (Pearson, r = 0.718, df = 11, p-value = 0.008), with conditions of accumulation (BCF≥1) in BhC where an average concentration between both climatic periods was observed in the seston of 0.027 ± 0.003 µg/g Hgd.w. (n = 6) and in the oyster of 0.144 ± 0.012 µg/g Hgd.w. (n = 36). While in CGSM hyperaccumulation conditions (BCF≥10) were reached with an average content in the seston of 0.007 ± 0.001 µg/g Hgd.w. (n = 6) and in the oyster of 0.074 ± 0.005 µg/g Hgd.w. (n = 35) (Table 1, Table S4).

Differences in Hg bioconcentration between CGSM and BhC were determined (Permanova, p-value < 0.05; Table 2). These differences were also observed as a function of climatic seasons, with an increase during the dry season in each CGSM season, and significant in the CIS-2 season in BhC compared to the rainy season. When the factors ecosystem and climatic season were combined, significant differences were still present, with higher values of Hg bioconcentration in CGSM in both climatic seasons compared to BhC (Table 1, Fig. 3B, Table S5). These results indicate that the oyster in CGSM is accumulating higher concentrations of Hg in its tissues compared to the BhC oyster, although the accumulation is also considerably higher in the BhC oyster.

Significant differences between juvenile and adult ages were determined with the Hg BCF, which was maintained when considering the climatic season (Table 2). In BhC, the highest values of Hg BCF were observed in juvenile ages in both climatic seasons. In CGSM, the highest BCF occurred in adult ages during the dry season, while they were similar in both ages during the rainy season (Spearman, r = 0.25, p-value > 0.05; Table 1, Fig. 3B, Table S4).

Relationship between environmental variables and Hg bioconcentration in oysters

Between the Hg concentration in the mangrove oyster tissue and its BCF by oyster in relation to the metal content in the seston, it was not possible to find a positive or negative relationship with the environmental variables analyzed. The relationship between physicochemical variables with Hg concentration and BCF in the mangrove oyster were not significant (DistLM; p-value > 0.05; Table 3). This suggests that environmental variables did not play a determining role in the differences in oyster Hg content and bioconcentration at CGSM and BhC (Fig. 3). Other factors, such as Hg content in the seston and local transport and sedimentation processes, may be playing a more influential role in Hg accumulation.

Table 3 Analysis DistLM in Hg concentration and bioconcentration factor in Crassostrea rhizophorae with physicochemical variables.

Analysis of DistLM in the relationship between Hg concentration and its bioconcentration in Crassostrea rhizophorae with physicochemical variables (predictors). Stepwise model and selection criterion of adjusted R2 were used (9,999 permutations). SS, sum of squares.

Variable	SS	Pseudo-F	p -value	Proportion of variation explained	
Juvenile size	
Temperature	11.2	0.41	0.54	0.039	
Salinity	28.1	1.09	0.33	0.098	
pH	27.8	1.07	0.33	0.097	
Dissolved oxygen	22.1	0.84	0.38	0.077	
Organic matter	0.61	0.02	0.89	0.021	
Adult size	
Temperature	1.2	0.03	0.88	0.026	
Salinity	4.5	0.09	0.77	0.098	
pH	39.9	0.95	0.36	0.087	
Dissolved oxygen	106.7	3.01	0.11	0.231	
Organic matter	0.1	0.03	0.96	0.026	
Notes.

p-values < 0.05 express that the variable significantly explains the variations in Hg content and its bioconcentration in Crassostrea rhizophorae. In bold is the greatest variation explained.

Hg contamination status of bivalves in a global context during the last five decadas

Based on a global assessment of total mercury (Hg) contamination in bivalves over the past 50 years across 79 study sites, risk levels were analyzed using the adapted Nemerow Pollution Index (Pc) based on the provisional tolerable weekly intake (PTWI) of one µg Hg/kg body weight. The Ciénaga Grande de Santa Marta (CGSM) ecosystem, evaluated in 2021 and 2022 with the oyster C. rhizophorae, falls within Clade 4. Although this clade includes sites exceeding the very high-risk threshold (Pn >3), it represents the cluster with the lowest relative risk among the four clades. CGSM exhibited a Pc value of 4.77, which, despite surpassing the threshold, remains far below the highest global values. CGSM clustered near sites such as Salerno (Italy, 1987) and Daksa (Croatia, 1984) with the mussel Mytilus galloprovincialis, and Chiayi (Taiwan, 1973) with the oyster Magallana gigas, highlighting a grouping associated with moderate mercury contamination. It is important to note that CGSM exhibited a higher risk of contamination compared to sites such as Saldanha Bay (South Africa, 2016) with Choromytilus meridionalis and M. galloprovincialis, Galicia (Spain, 2006 and 2010) with the scallop Mimachlamys varia, and Changseon Coast (South Korea, 2008) with M. galloprovincialis, all of which remained below the very high-risk threshold with Pn values of 2.33, 2.75, and 2.89, respectively (Fig. 4, Tables S6 and S7).

Figure 4 Pollution index (Pc) due to Hg in bivalves from 1970 to 2022.

Hierarchical clustering analysis was conducted to evaluate the Nemerow pollution index (Pc) due to Hg in marine and coastal ecosystems measured in bivalves from 1970 to 2022. Belgium (BEL), Brazil (BRA), China (CHN), Colombia (COL), Croatia (HRV), Denmark (DNK), France (FRA), Indonesia (IDN), Italy (ITA), Montenegro (MNE), Netherlands (NLD), Norway (DNK), Portugal (POR), South Africa (ZAF), South Korea (KOR), Spain (SPA), Taiwan (TWN), United Kingdom (GBR), United States (USA), Vietnam (VNM). Clade 1, 2 and 3: sites with very high risk pollution. Clade 4: sites with high risk pollution to very high risk pollution. The arrow refers to the sites evaluated in the present study.

In contrast, the Cispatá Bay (BhC) ecosystem, assessed in 2021 and 2022 with C. rhizophorae, recorded a Pc value of 5.92 and was grouped within Clade 3, positioning it in the third-highest contamination cluster. BhC clustered with sites such as the Yellow Sea and Bohai Sea (China, 2019) with Mactra veneriformis, Chlamys farreri, Ruditapes philippinarum, M. gigas, and Cyclina sinensis; the Sibenik Aquarium (Croatia, 1983) and Piombino (Italy, 1987) also with M. galloprovincialis. This indicates a higher relative risk compared to CGSM, but still distant from the extreme contamination observed in Clade 1, which includes heavily impacted sites such as Kastela Bay (Croatia, 1980–1985) with M. galloprovincialis, the “Cheminova” factory area (Denmark, 1982–1983) with M. galloprovincialis and Macoma balthica, and the English Channel (United Kingdom, 1975) with Mytilus edulis. In these locations, Hg concentrations in bivalves exceeded the PTWI limit of one µg Hg/kg body weight established by the World Health Organization (WHO) by more than an order of magnitude (Fig. 4, Tables S6 and S7).

These results highlight that although both Colombian sites exceed the PTWI threshold and thus warrant attention, they are far from the global hotspots of mercury contamination in bivalves. Instead, they are grouped within clusters representing intermediate pollution levels when analyzed in a historical global context.

Discussion

The higher concentrations of Hg in sediments, seston, and oysters in BhC compared to CGSM (Fig. 2B, Table S1) highlight significant concerns regarding environmental quality and ecosystem health. The elevated Hg levels in BhC are primarily linked to the Sinú River’s input through the Sicará stream, which carries water and sediments contaminated by agricultural runoff (Campos, Dueñas Ramírez & Genes, 2015). Among these, the application of fungicides containing phenylmercury (C6H5Hg) and the extensive spraying of rice fields with agrochemicals rich in mercury-based compounds (Marrugo-Negrete et al., 2020) are particularly significant sources. These practices contribute to the continual release of Hg into the estuarine system, exacerbating contamination levels over time. Additional sources of Hg in BhC include regional artisanal and small-scale gold mining operations, discharges of untreated municipal and industrial wastewater, the historical use of Hg-based paints as anti-corrosion agents on ships, and atmospheric deposition from regional emissions (Burgos-Nuñez et al., 2014; Burgos-Nuñez et al., 2017).

In CGSM, the sources of Hg contamination are less well-defined. However, the entry of Hg into this system is associated with atmospheric deposition and anthropogenic activities, such as industrial discharges and gold mining in areas connected to the Magdalena River (Alonso et al., 2000; Mancera-Rodríguez & Álvarez-León, 2006). Given CGSM’s status as a Ramsar Wetland and Biosphere Reserve, identifying and mitigating potential contamination sources is crucial to preserving its ecological integrity and the socioeconomic benefits it provides to local communities.

While Hg levels in sediments in BhC and CGSM remain below the tolerable ecological threshold of 0.13 µg/g Hgd.w. (TEL) (Buchman, 2008), they are significantly lower than those observed in more heavily impacted regions, such as Cartagena Bay, Colombia (0.094–10.293 µg/g Hgd.w.) (Alonso et al., 2000), and San Vicente Bay, Chile (0.37–0.95 µg/g Hgd.w.) (Díaz et al., 2001). However, it is important to note that BhC exhibits a higher contamination risk compared to CGSM, as previous assessments have reported Hg concentrations exceeding the TEL threshold in sediments along the Sinú River and near the mouth of BhC by Feria, Marrugo & González (2010), Campos, Dueñas Ramírez & Genes (2015), and Marrugo-Negrete et al. (2020) reported Hg concentrations exceeding the TEL threshold in sediments along the Sinú riverbed and at the mouth of BhC. In contrast, in CGSM, Hg concentrations in sediments have remained below 0.11 µg/g Hgd.w., similar to the findings of this study (Fig. 2B, Table S4).

The slight increase in Hg content in sediment and seston during the rainy season compared to the dry season (Fig. 2B), may be attributed to increased metal transport from terrestrial sources. Rainfall-induced sediment flushing and freshwater inflow during this period mobilize Hg from upstream sources into the estuarine systems, highlighting the critical role of hydrological dynamics in shaping contamination patterns (Da Silva Ferreira et al., 2013). This phenomenon is particularly pronounced in BhC, where contributions from the Sinú River amplify Hg loading (Marrugo-Negrete et al., 2020). Similarly, in CGSM, the Magdalena River serves as a major pathway for metal transport, emphasizing the interconnectedness of terrestrial and aquatic systems in driving contamination processes (Mancera-Rodríguez & Álvarez-León, 2006; Table 3, Fig. 3A).

Additionally, although the water column Hg concentration was slightly higher in BhC (0.005 × 10−1 ± 0.003 × 10−2 µg/L) similar to what was observed in the area by Marrugo-Negrete & Paternina-Uribe (2011) than in CGSM (0.003 × 10−1 ± 0.009 × 10−2 µg/L), these values are comparable to those reported in coastal systems such as Hawaii, USA (Hunter et al., 1995), Tongyeong, South Korea (Mok et al., 2015), and the Tianjin Reservoir in China (Zhang et al., 2016). In contrast, significantly higher concentrations exceeding one µg/L have been recorded in heavily impacted areas, including 11 coastal sites in Italy (Berhnard, 1998), Chao Phraya estuary in Thailand (Hungspreugs et al., 1989), and the Adriatic Sea in Croatia (Kosta et al., 1978), where mercury contamination has been attributed to a combination of natural geological sources and anthropogenic inputs such as industrial effluents and mining activities (Tables S8 and S9). These comparisons emphasize that even moderate elevations in dissolved Hg, such as those observed in BhC, may enhance Hg uptake in filter-feeding organisms. This could explain the higher mercury concentrations detected in oyster tissues from BhC, despite presenting lower a bioconcentration factor (BCF) compared to CGSM (Table 1). These results highlight the importance of the sediment-seston interaction and local conditions in influencing Hg availability for oysters in the coastal areas of the Colombian Caribbean (Aguirre-Rubí et al., 2017).

Environmental factors, including temperature, organic matter and silt content, appear to play a pivotal role in influencing Hg concentrations in sediments and seston. In BhC, significant correlations were observed between elevated Hg concentrations in seston and higher water temperatures during the rainy season (Fig. 2). This relationship aligns with findings that temperature accelerates chemical reaction rates, such as the methylation of Hg, thereby increasing its bioavailability (Richard et al., 2016). Additionally, higher Hg concentrations in sediments were associated with increased organic matter and silt content during the rainy season, emphasizing the role of organic matter in retaining metals in sediments, particularly in fine sediments and sulfate-reducing environments (Cogua, Campos-Campos & Duque, 2012), as observed in both CGSM (Espinosa, Parra & Villamil, 2011) and BhC (Fig. 2A, Table S1).

Although no direct correlation was identified between pH and Hg concentrations in sediments and seston (Table S3), slightly acidic to neutral pH conditions are known to enhance Hg precipitation in sediments (Parra & Espinosa, 2008). This mechanism likely contributes to the observed higher Hg concentrations in BhC sediments, which exhibited lower pH values compared to CGSM (Fig. 2). Furthermore, seasonal fluctuations in pH, particularly during the transition between rainy and dry seasons, may influence the solubility and availability of Hg, adding another layer of complexity to its distribution patterns.

Additionally, variations in sulfate to sulfide conversion processes may increase the flux of reactive phosphate and ammonium at the sediment-water interface (Uwah et al., 2013). This process favors the precipitation of Hg in sediments as insoluble oxides, carbonates, or phosphates (Volety, 2008; Azizi et al., 2018a). The interaction of these processes could account for the observed variations in Hg content in sediments, especially when considering seasonal pH fluctuations. This pattern is particularly evident in BhC due to the pH differences between the rainy and dry seasons (Fig. 2).

The role of salinity in modulating Hg methylation and sediment retention also warrants attention. Elevated salinity levels, such as those observed in BhC during the dry season and at CGS-3 in CGSM, can inhibit Hg2+ methylation through the production of hydrogen sulfide (H2S). This compound forms mercury sulfide (HgS), a mineral that is poorly available for methylation processes, thereby limiting the bioavailability of Hg (Compeau & Bartha, 1984). These findings highlight the intricate interplay of physicochemical factors in shaping Hg dynamics in estuarine ecosystems.

Regarding the mangrove oyster, the influence of environmental variables on Hg concentration and bioconcentration factor (BCF) was not significant (Tables 1 and 2, Fig. 3). This aligns with findings from other studies where the effects of variables such as temperature, salinity, and pH on Hg uptake and accumulation in bivalves remain complex and not fully understood (Volety, 2008), unlike the clearer dynamics of Hg in sediments and seston, where processes such as accumulation, uptake, toxicity, and speciation are well-documented (Suryanto Hertika et al., 2021), the factors influencing Hg bioconcentration in oysters are less understood and may be influenced by the organism’s unique physiological and ecological traits.

Despite the lack of significant correlations, it is worth considering the potential role of high dissolved oxygen concentrations observed in CGSM. Elevated oxygen levels, coupled with variations in sediment chemical composition, can influence the metabolic activity of bivalves, potentially altering their ability to absorb and excrete Hg (Silva, Rainbow & Smith, 2003). These findings suggest that while direct correlations may not be evident, indirect effects mediated through environmental and physiological interactions could still play a role in Hg bioconcentration.

Mangrove oysters are known for their ability to filter large volumes of water during feeding (Restrepo & López, 2008), capturing particulate matter including seston-bound metals (Coimbra, 2003). Accordingly, Hg concentrations in oysters are closely related to the metal content in seston, as evidenced in this study (Fig. 2B). Hg bioconcentration was significantly associated with seston, which is expected given that oyster collection was primarily from mangrove roots submerged to depths greater than half a meter. As with seston, metal accumulation and hyperaccumulation were observed in relation to Hg concentrations in sediment (Tables 1 and 2). These findings underscore the importance of measuring metal concentrations in sediments and seston to assess Hg availability and uptake by bivalves, providing a comprehensive understanding of the interaction between these organisms and their contaminated environments.

The findings of Bayne et al. (1989) provide critical insight into this relationship, demonstrating that feeding, digestion, and growth in M. edulis are strongly influenced by seston concentration and particulate organic matter (POM) content. At higher seston levels, ingestion rates increase until a physiological limit is reached, which the production of pseudofaeces, limiting net absorption. These thresholds of feeding efficiency suggest that oysters in seston-rich environments, such as BhC during the rainy season (Fig. 2B, Table 1), may experience both higher ingestion and variable assimilation of metal-laden particles, thereby modulating Hg accumulation.

In parallel, DeForest, Brix & Adams (2007) emphasize that bioaccumulation factors (BAFs) and bioconcentration factors (BCFs) for metals often exhibit inverse relationships with exposure concentrations. This phenomenon arises from regulatory mechanisms in aquatic organisms, such as metal storage, excretion, and physiological limits of uptake. For Hg, this inverse trend means that elevated environmental concentrations may not linearly correspond to tissue accumulation, complicating risk assessments.

Similarly, Chernova & Shulkin (2019) highlight the variability of BCFs and BAFs in natural environments, especially under low to moderate metal concentrations. Their work with marine algae illustrates that under background or fluctuating concentrations, metal uptake can show high variability due to factors like organic matter binding, hydrodynamics, and physiological state of organisms. This variability underscores the importance of interpreting BCF values within the context of both exposure intensity and organismal condition.

Taken together, these findings reinforce that Hg accumulation in C. rhizophorae is not merely a function of dissolved concentrations but is also significantly modulated by seston dynamics, feeding physiology, and the oyster’s regulatory mechanisms. These insights advocate for integrated monitoring approaches that include environmental variables (e.g., seston load, POM quality), organismal traits (e.g., filtration capacity), and spatial–temporal exposure patterns to accurately assess metal transfer and risk in estuarine systems.

Interestingly, the bioconcentration patterns observed in BhC, where adult oysters showed lower Hg concentrations than juvenile oysters (Fig. 3B), align with previous studies. For example, Coimbra (2003) in Sepetiba Bay, Brazil, with Mytela guyanensis and Díaz et al. (2001) in San Vicente Bay, Chile, with Tagelus dombeii, reported inverse correlations between Hg content and species age. They suggested that metal assimilation rates decrease as the excretion rate increases in larger individuals, likely due to reduced metabolism and less water filtration with bivalve growth (Azizi et al., 2018b).

Several mechanisms regulate the accumulation of toxic metals such as Hg in bivalve tissues during their growth. One such mechanism is the formation of mineralized granules, which allows for Hg storage and potential detoxification (Cossa, 1989). Bivalves also regulate Hg concentrations through excretion mechanisms via urine and feces, maintaining appropriate Hg levels (El-Moselhy & Yassien, 2005). The development of new gill systems in bivalves plays a crucial role in filtering particles, including metals, from the aquatic environment (Kumar Gupta & Singh, 2011). As bivalves grow, this gill development enhances their ability to capture and regulate Hg in their tissues.

Another important strategy for mitigating the accumulation of toxic metals in bivalves is the release of gametes during reproduction (Cossa, 1989). In both CGSM and BhC, the highest BCF values in oysters were observed during the dry season (Table 2, Fig. 3B). During gamete release, which typically occurs in the rainy season and includes several reproductive peaks in the Colombian Caribbean (López-Sánchez & Mancera-Pineda, 2019), mineralized granules stored in lysosomes may be expelled along with the gametes (Costa, Paiva & Moreira, 2000). This exocytosis process releases lysosomal contents, including metals like Hg, into the aquatic environment (Cossa, 1989). This possible release of Hg granules during gametogenesis may explain the lower BCF values observed in the rainy season.

Conversely, the highest BCF values were observed during the dry season, particularly in CGSM, where adult oysters exhibited greater Hg concentrations and BCF values than juveniles. These findings are consistent with previous research by Costa, Paiva & Moreira (2000), who documented similar trends in other estuarine ecosystems. The contrasting patterns between CGSM and BhC highlight the complexity of environmental and organismal factors that influence Hg bioconcentration and bioconcentration dynamics.

Although environmental variables such as temperature, salinity, pH, and dissolved oxygen were initially considered as potential drivers of mercury bioconcentration in oysters, the results indicate that the primary factor influencing Hg accumulation was the mercury content in seston, rather than direct physicochemical conditions of the water column.

Hg intake risk in C. rhizophorae at CGSM and BhC compared to global hotspots

This study offers a critical contribution to the understanding of total Hg pollution in C. rhizophorae within the Colombian Caribbean, specifically in the ecosystems of Ciénaga Grande de Santa Marta (CGSM) and Cispatá Bay (BhC). Among 79 study sites spanning 36 bivalve species and 32 global investigations from 1970 to 2022 (Fig. 4, Table S6), CGSM and BhC are the only locations where C. rhizophorae has been assessed for Hg contamination using wet weight concentrations (µg/kg). These data are standardized against the provisional tolerable weekly intake (PTWI) of one µg Hg/kg body weight established by FAO/WHO (2010), allowing for robust and consistent comparison of Hg exposure risk through seafood consumption.

Although Hg concentrations in C. rhizophorae from CGSM (16.7 ± 1.9 µg/kgw.w., n = 35) and BhC (29.2 ± 3.2 µg/kgw.w., n = 36) exceeded the PTWI threshold, these levels remain significantly lower than those found in ecosystems classified in Clade 1, which includes historical global hotspots of extreme contamination. For instance, M. galloprovincialis from Kastela Bay (Croatia) reached 9,525 ± 2,337 µg/kgw.w. (Mikac et al., 1985), while M. balthica near the “Cheminova” factory (Denmark) recorded 1,260.3 ± 119.6 µg/kgw.w. (Riisgard et al., 1985). Other examples include S. plana from the Ria de Aveiro (Portugal) with 395 ± 87 µg/kgw.w. (Coelho et al., 2014), and Tegillarca granosa from East Java, Indonesia with 272.5 ± 172.1 µg/kgw.w. (Soegianto et al., 2020) (Fig. 4, Tables S6 and S7).

While these international assessments have traditionally emphasized species like M. edulis and M. galloprovincialis monitored in more than 10 countries such as Italy (Brambilla et al., 2013; Esposito et al., 2021), Spain (Olmedo et al., 2013), and South Africa (Firth et al., 2019), the representation of C. rhizophorae in global or regional Hg monitoring is notably scarce. Our study fills this critical gap by generating the first standardized dataset for this ecologically and commercially important Caribbean species, evaluated through the lens of direct human exposure risk. In fact, most historical studies on C. rhizophorae have used dry weight units, as seen Dominican Republic (Sbriz et al., 1998), Brazil (Silva, Rainbow & Smith, 2003), and the Nicaragua (Aguirre-Rubí et al., 2017), which complicates direct risk assessment for consumption compared to the approach used here.

This highlights not only the regional novelty of the current study but also the broader need for consistent wet weight-based assessments of Hg in seafood species of local significance. Moreover, by implementing an adapted Nemerow Pollution Index standardized for PTWI, this study advances a globally comparable metric of consumer health risk one that remains absent from many other international assessments.

Nevertheless, the current state of Hg contamination in Colombia, particularly in Cartagena Bay, highlights the need for vigilance. Cartagena Bay has recorded some of the highest global Hg contamination risks for bivalves in the past decade (Aguirre-Rubí et al., 2017), attributed to historical discharges from industrial facilities like the Alcalis chlorine plant (Mancera-Rodríguez & Álvarez-León, 2006; Bolaños-Alvarez et al., 2024; Ucros-Rodríguez et al., 2025). Similar trends have been observed in other regions, including “Cheminova” factory, Denmark (Riisgard et al., 1985), Kastela Bay, Croatia (Mikac et al., 1985; Martinčić et al., 1987) San Vicente Bay, Chile (Díaz et al., 2001), California Gulf, United Stades (Ruelas-Inzunza, Soto & Páez-Osuna, 2003), and the Adriatic Sea, Croatia (Kljaković-Gašpić et al., 2010), reinforcing the global relevance of monitoring mercury in estuarine and coastal ecosystems. In this context, the current data from CGSM and BhC establish vital benchmarks that can inform targeted mitigation strategies, ensuring ecosystem protection and seafood safety in the region.

Conclusions

Cispatá Bay (BhC) presented higher concentrations of Hg in sediments, seston, and oysters compared to Ciénaga Grande de Santa Marta (CGSM) across both climatic seasons. However, CGSM oysters showed greater bioconcentration factors (BCF), particularly in relation to seston, suggesting a higher capacity for Hg accumulation in this ecosystem. Temperature in the water column and organic matter in sediments significantly influenced Hg concentrations in seston and sediments but showed no significant relationship with Hg bioconcentration in the oysters. Adult oysters accumulated more Hg in CGSM, whereas juvenile oysters accumulated more Hg in BhC, underscoring the importance of considering the bivalve age when assessing Hg contamination in different ecosystems. When assessed using the provisional tolerable weekly intake (PTWI) of one µg Hg/kg body weight as a health risk threshold, both ecosystems exceeded the recommended limit, indicating a potential risk from oyster consumption. However, the contamination index analysis places CGSM and BhC far from the most critical pollution hotspots on a global scale. These findings emphasize the importance of continued monitoring of Hg in estuarine systems, considering both environmental dynamics and biological factors that modulate metal accumulation.

Supplemental Information

Supplemental Information 1 Results of physicochemical variables

Raw data in the results of physicochemical variables measured at a depth of 0.5 m in the water column, and percentage of organic matter, redox potential and silt content in the sediment.

Supplemental Information 2 Hg content in the environment

Raw data in the results of Hg concentrations in µg/g in sediments (dry weight) and seston (dry weight and wet weight) in the Ciénaga Grande de Santa Marta (CGSM), and Cispatá Bay during the rainy season (November 2021) and dry season (March 2022).

Supplemental Information 3 Correlations in the Hg content in sediments and seston with respect to the environmental variables

Results of correlations in the Hg content in sediments and seston with respect to the environmental variables measured in the stations of Ciénaga Grande de Santa Marta (CGSM) and Cispatá Bay (BhC) during the rainy season (November 2021) and dry season (March-April 2022). Red font color denotes high and significant correlations.

Supplemental Information 4 Hg content and anteroposterior length Crassostrea rhizophorae

Raw data in the results of Hg concentrations in µg/g dry weight (d.w.), wet weight (w.w.), and anteroposterior length (APL) average (avg) and desviation (desv) in the oysters Crassostrea rhizophorae. Ciénaga Grande de Santa Marta (CGSM) and Cispatá Bay (BhC).

Supplemental Information 5 PERMANOVA comparisons on Hg content and bioconcentration factor in Crassostrea rhizophorae

PERMANOVA comparisons were conducted on mercury concentrations in µg/g dry weight (d.w.) and its bioconcentration factor (BCF) vs. ages (juveniles and adults), stations, ecosystem, and climatic season in the oyster Crassostrea rhizophorae. Ciénaga Grande de Santa Marta (CGSM) and Cispatá Bay (BhC).

Supplemental Information 6 Record of the concentration and contamination level in bivalves from 2010 to 2022

Record of the concentration and contamination level by the Nemerow Comprehensive Pollution Index (Pc) of Hg in marine-coastal ecosystems in Crassostrea rhizophorae and among all bivalve species in the time season from 1970 to 2022. Pollution level: 2.0 < Pc ≤ 3.0 high ∘, and Pc > 3.0 very high ∘. Avg: average, EE: standard error, min: minimum y max: maximum.

Supplemental Information 7 Record of the Hg concentration in bivalves from 1970 to 2022

Raw data in the record of the Hg concentration in marine-coastal ecosystems in Crassostrea rhizophorae and among all bivalve species in the time season from 1970 to 2022, for evaluated the pollution index (Pc).

Supplemental Information 8 Hg content in the water column at different coastal sites around the world

Supplemental Information 9 Raw data Hg content in the water column

Raw data in the record of the Hg content in the water column at different coastal sites around the world.

We would like to especially thank the members of the team Diana Bustos-Montes, Paulo Tigreros-Benavides, Diana Rubio-Lancheros, María Camila Castellanos-Jimenez, Ana María Hernández-Chamorro, Nicólas Santos-Vásquez, Andrés Navarro-Martínez, Nelson Rafael Camargo-Tibamoso, Laura Daniela García-Meléndez and the students of the Marine Ecology and Biodiversity (ECOBIOMAR-UTADEO) research incubator, for their support in taking samples and laboratory analysis. Special thanks to CECIMAR of the Caribbean headquarters of the National University of Colombia for providing their spaces for sample processing. To INVEMAR for the support provided for the treatment of samples.

Additional Information and Declarations

Competing Interests

Author Contributions

Field Study Permissions

Data Availability

The authors declare there are no competing interests.

Anubis Vélez-Mendoza conceived and designed the experiments, performed the experiments, analyzed the data, prepared figures and/or tables, authored or reviewed drafts of the article, and approved the final draft.

Jeimmy Paola Rico-Mora conceived and designed the experiments, performed the experiments, analyzed the data, authored or reviewed drafts of the article, and approved the final draft.

Néstor Hernando Campos-Campos conceived and designed the experiments, performed the experiments, analyzed the data, authored or reviewed drafts of the article, and approved the final draft.

Margui Lorena Almario-García conceived and designed the experiments, authored or reviewed drafts of the article, and approved the final draft.

Adolfo Sanjuan-Muñoz conceived and designed the experiments, authored or reviewed drafts of the article, and approved the final draft.

The following information was supplied relating to field study approvals (i.e., approving body and any reference numbers):

277 individual oysters were collected from CGSM, and 237 from BhC under the collection permit for wild species specimens of biological diversity for non-commercial scientific research purposes, granted by the Autoridad Nacional de Licencias Ambientales (ANLA) through Resolution 1271 of October 23th 2014, modified by resolutions 1715 of December 30th 2015 and 00213 of January 28th 2021, to the University of Bogota Jorge Tadeo Lozano (UTADEO).

The following information was supplied regarding data availability:

Raw data is available in the Supplemental Files.

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
