# Peer review of "Changes in mercury content in oysters in relation to sediment and seston content in the Colombian Caribbean lagoons"

_PeerJ, doi:10.7717/peerj.19868_

## Round 0.1 · original submission · Major Revisions

Dear Dr. Vélez-Mendoza,

We are pleased to inform you that your manuscript has passed the peer review stage and is ready for revision.

The reviewers provided detailed comments, and I ask that you consider these carefully when revising the manuscript as well as respond to their suggestions in the cover letter when you re-submit. This will help avoid further rounds of explanations and revisions, and allow quickly move to the decision.

·

Basic reporting

The reviewer is not a native speaker, so cannot assess the quality of the language.
Introduction and background correspond to the context. References are relevant.

Structure conforms to PeerJ standards, however, the following should be added - Additional information and declarations: Funding; Grant Disclosures; Competing Interests; Author Contributions; Data Availability

Figures are relevant, of high quality, well labelled and described. Raw data are supplied.

Experimental design

Original primary research is within the scope of the journal.
Research question is well defined, relevant and meaningful.
The study provides new data on the mercury content in the mangrove oyster Crassostrea rhizophorae, in sediments, and in the seston from the Ciénaga Grande de Santa Marta (CGSM) and Cispatá Bay (BhC) during two climatic seasons (rainy and dry ones).

Rigorous investigation corresponds to a high technical and ethical standard.

Methods are described with sufficient detail except some methodological incorrectness.
In all the studied objects (sediments, seiston and oysters), some mercury may be present in volatile form (organic mercury, e.g. Hg(CH3)2, Hg(C2H5)2, which is lost during thermal treatment of the samples (the authors used heating to 450-500 0C during sample preparation). In particular (especially), microorganisms that form volatile organic forms of this metal are present in the sediments. These forms can also be formed in biota, i.e. seiston and oysters. Therefore, it is necessary to clarify whether the volatile form of mercury was captured. If not, it should be clarified in the methodology.

Validity of the findings

All underlying data have been provided; they are robust, statistically sound, and controlled.
Conclusions require further elaboration
Lines 593-596 need to be rewritten (indicated below)

Additional comments

Thanks to the authors for the opportunity to review the results of the research.
The paper is devoted to the accumulation of mercury in oyster from two water areas differing in the degree of pollution. Particularly interesting are the results obtained by the authors on the bioconcentration of mercury in the food chain in different seasons.
The authors investigated the bioconcentration (when it is being compared to the external environment, i.e. seawater) or mercury content in the samples. Bioaccumulation is the change in mercury concentration over time. We therefore strongly recommend to replace the term ‘bioaccumulation’ with ‘bioconcentration’.
The issue of mercury accumulation by differently aged molluscs has always been controversial, which is not clearly understood in this paper and requires further investigation. In addition, it is more appropriate to rely on age rather than mollusk size to assess the age-related changes in mercury content (as growth rates may vary under different conditions).
The paper presents the comparison on of the mercury content in the oyster with the acceptable limit for human consumption, but the value of the limit of 0.5 µg/g body weight is not clear. Is this the value of mercury intake in an oyster per day, week, or month? Over what period of time? Therefore the conclusion that mercury levels in oysters from the conditionally clean site remained below the permissible limit is not clear. It is not also clear what risk is being discussed, contamination risk or consumption risk? Lines 593-596 should be rewritten.

Line 199-201 presents the method of atomic absorption spectrometry, but the reference to the source of the citation by Fernandez-Martinez et al., 2015 - atomic fluorescence spectrometry
Line 256 should indicate that the formula for calculating the Nemerow integral contamination index (Pc) has been modified
Table 1 and line 222 (BCFsd and BCFst) should be aligned.
Line 474-476: The word ‘hydroxides’ should be deleted as mercury hydroxides Hg+OH- and Hg2+(OH)2- are not stables, and do not exist in aqueous medium, they decompose.

Reviewer 2 ·

Basic reporting

The study meets all the required characteristics in terms of structure and content. However, it does not include enough studies in the introduction to support the mercury issue and the hypothesis. Additionally, it is necessary to include more recent references.

Experimental design

'no comment'

Validity of the findings

'no comment'

Additional comments

I recommend incorporating more recent studies in the Introduction and Discussion sections.

·

Basic reporting

No comment

Experimental design

No comment

Validity of the findings

No comment

Additional comments

Overall, this is a very interesting study for readers. The data can be a good contribution to the scientific literature. The presentation of the manuscript needs minor revisions. Some changes/suggestions are required before publishing in a worthy journal, therefore, my recommendation is to accept.

·

Basic reporting

The authors use clear and unambiguous professional English. The text is easy to read by a foreigner.
The authors use modern literature. Fundamental works on this topic were written in the 20th century. The authors haven't read them enough.
The professional structure of the article, figures, tables is good. The initial data is published. But I propose to provide data on the concentration of seston in the water (mg/l) and on the granulometric content (content of silt) of bottom sediments as parameters of envoronment.

The results do not match the hypothesis of the pollution of the areas.
The conclusion is general, as the authors do not believe their own data on relatively low concentrations of mercury in sediments and oysters, and due to an error in calculating the Nemerov integral contamination Index, they received an increased Index than other authors.

Experimental design

The methods are described with sufficient detail and information to reproduce, but there are shortcomings:
Field phase
That chapter has the information about of preparing of sediments and oysters, but don’t has information about preparing of seston
Lines 152-157. Seston was not determinated in mg/l
line 159 there is no of beginning of sentence.
Laboratory phase
Lines 178 I don’t understand what “all materials” were pre-treated by HNO3. Samples of oyster? BS? Seston? May be chemical tableware?
Line 184 “The anteroposterior length (APL) was measured on………” ore “The anteroposterior length (APL) of oyster was measured on….”
Line 197 Chapter don’t have the information about granulometric analysis of sediment and seston, but has information that “Sediment and seston fractions smaller than 65 µm were digested with 5% nitric acid”. Information about preparing of seston must add to Laboratory phase.
Data analysis
The authors made a mistake, and therefore part of the work should be recalculated and rewritten:
the authors used the threshold concentration of mercury in oysters calculated for the wet mass, while they used concentrations expressed for the dry mass. As a result, the Nemerov integral contamination Index was overestimated by ~5 times. If the authors recalculated their data on the raw mass, they did not report it anywhere.

Validity of the findings

– The disadvantages are – the threshold concentration of Hg (in shellfish used in food) - 0.5 µg/g wet masses – I do not recommend using it as a criterion for environmental assessment. It was created for sanitary purposes for food facilities.
– the authors did not take into account that BCF (seston-oysters) can increase in uncontaminated conditions (DeForest et al., 2007, Chernova, Shulkin, 2019). And low concentrations of the element in the ecosystem (in food, water) lead to an increase in the rate of physico-chemical processes of accumulation and sorption of elements.
– the authors did not calculate the concentration of seston in water, which is important for the transfer of mercury along the seston-shellfish food chain. It is known that with an increase in the concentration of seston in water (mg/l), the filtration rate of shellfish decreases. Suspended OM (microorganisms, phytoplankton, detritus) is food for filter-feeding shellfish, and the inorganic matter of seston is selectively filtered out by shellfish (about 50% -Bayne 1989), it does not enter the digestive tract. I recommend determining the amount of organic matter concentration in seston in future studies.
– The BCF calculation of shellfish-bottom sediments should be interpreted with caution. This value has no physical meaning. Shellfish do not use BS for food, they are 0.5 m away from the surface of BS. Only thin sediments can be stirred. The authors could have specified the composition and quantity of the silty fraction, but did not report it.
– The authors did not take into account the background mercury concentrations in sediments and in oysters and in seston too. Background concentrations of Hg in shellfish may have systematic differences. Mercury, a chemical element common in nature, has a natural background in all components, which should not be considered pollution.
GENERAL REMARKS
The experimental part of the article is well done, with the using of modern technologies, results are new, interesting and useful, but article contents needs serious revision and re-review.

Additional comments

For authors:
Page 30: Fig. 3. Comparative analysis of concentrations (A) and bioconcentration factor (B) of Hg in the oyster Crassostrea rhizophorae between Ciénaga Grande de Santa Marta (CGSM) and Cispatá Bay (BhC): effects of organism size and climatic season. Above the green line, Hg accumulation condition in the oyster is considered (FBC>1), and above the red line, hyperaccumulation of the metal is considered (FBC>10). I don’t understand what is FBC? Is it mistake?

Bayne B. L. Effects of seston concentration on feeding, digestion and growth in the mussel Mytilus edulis / B. L. Bayne, A. J. S. Hawkins, E. Navarro, I. P. Iglesias // Marine Ecology – Progress Series. – 1989. – Vol. 55, is. 1. – P. 47–54.
Chernova E. N. Concentrations of Metals in the Environment and in Algae: The Bioaccumulation Factor / E. N. Chernova, V. M. Shulkin // Russian Journal of Marine Biology – 2019. – Vol. 45, № 3. – P. 191–201. – DOI: 10.1134/S1063074019030027.
DeForest D. K. Assessing metal bioaccumulation in aquatic environments: The inverse relationship between bioaccumulation factors, trophic transfer factors and exposure concentration / D. K. DeForest, K. V. Brix, W. J. Adams // Aquatic Toxicology. – 2007. – Vol. 84, is. 2. – P. 236–246.

---

## Round 0.2 · accepted · Accept

In the revised version the authors took into consideration all comments and remarks. I recommend to accept the manuscript for publication in PeerJ.